# Melatonin at the Crossroads of Oxidative Stress, Immunity, and Cancer Therapy

**DOI:** 10.3390/antiox15010064

**Published:** 2026-01-03

**Authors:** Elena Lavado-Fernández, Cristina Pérez-Montes, Miguel Robles-García, Adrián Santos-Ledo, Marina García-Macia

**Affiliations:** 1Institute of Functional Biology and Genomics (IBFG), University of Salamanca/CSIC, 37007 Salamanca, Spain; ellavadof@usal.es; 2Department of Biochemistry and Molecular Biology, Universidad de Salamanca, 37007 Salamanca, Spain; 3Institute of Neuroscience of Castilla y Leon (INCyL), 37007 Salamanca, Spain; cristina_perez_montes@usal.es; 4Department of Human Anatomy and Histology, Universidad de Salamanca, 37008 Salamanca, Spain; mroblesgarcia@usal.es; 5Department of Statistics, Universidad de Salamanca, 37007 Salamanca, Spain

**Keywords:** melatonin, oxidative stress, antioxidants, immune modulation, macrophages, tumor microenvironment, cancer therapy

## Abstract

Melatonin, an ancient and evolutionarily conserved indolamine, has long attracted attention for its multifunctional roles in redox homeostasis. More recently, it has been studied in relation to immune regulation and cancer biology. Beyond its well-known circadian function, melatonin modulates oxidative stress by directly scavenging reactive oxygen and nitrogen species and by upregulating antioxidant enzymes, including superoxide dismutase, catalase, and glutathione peroxidase. At the same time, it exerts wide-ranging immunomodulatory functions by influencing both innate and adaptive immune responses. All these actions converge within the tumor microenvironment, where oxidative stress and immune suppression drive cancer progression. Although the antitumoral effects of melatonin have traditionally been interpreted through its actions on T cells and NK cells, recent studies identify macrophages as an underappreciated and pivotal target. Notably, melatonin influences macrophage polarization, favoring antitumor M1 phenotypes over pro-tumoral M2 states, while attenuating chronic inflammation and restoring mitochondrial function. This review summarizes current knowledge on melatonin’s antioxidant and immunoregulatory mechanisms, highlighting its impact on the tumor immune microenvironment, with a particular focus on the growing recognition of macrophages as a compelling new axis through which melatonin may exert anticancer effects.

## 1. Introduction

Melatonin, or N-acetyl-5-methoxytryptamine, is a small indolamine with numerous physiological and biological roles [1]. Aaron Lerner and colleagues first isolated melatonin from bovine pineal glands in 1958 [2,3,4]. They named it melatonin because these pineal gland extracts contained a molecule capable of lightening frog and fish melanocytes [5]. Melatonin production is found across all domains of life, including organisms without a pineal gland such as archaea, bacteria, unicellular eukaryotes, invertebrates, and even vascular plants [6]. Several studies suggest that mitochondria serve as the primary site of melatonin synthesis, supporting its production by phylogenetically distant species [5,6,7]. It has been reported that oocyte mitochondria synthesize their own melatonin [8], which supports the idea that mitochondria in adult mammalian cells may also produce melatonin [9]. Other studies have shown that melatonin in the brain is synthesized by neuronal mitochondria. Suofu et al. demonstrated that melatonin synthesis is tightly linked to mitochondrial activity at least in the brain [10]. In plants, additional evidence indicates that chloroplasts also contribute to melatonin biosynthesis. The precise synthetic pathways in microorganisms remain largely unknown, although recent studies have shed some light on this area [11,12,13]. In humans, melatonin is synthesized in almost all cell types, tissues, and organs. The pineal gland is the most well-known site of production; however, melatonin is also produced in other tissues, such as enterochromaffin cells of the gastrointestinal tract and skin cells [5]. Melatonin derives from tryptophan through a process of hydroxylation by the enzyme tryptophan hydroxylase [1,14,15]. The resulting product, 5-hydroxytryptophan, is decarboxylated by the enzyme aromatic amino acid decarboxylase resulting in serotonin. Serotonin will be transformed into melatonin by the sequential action of the enzymes N-acetyltransferase and N-acetylserotonin O-methyltransferase [14,15]. Once synthesized, this hormone is released into the systemic circulation, allowing it to reach all tissues [16]. Melatonin endogenous levels are mainly suppressed by light exposure during the dark time (night) [17], and aging also causes a natural decline of melatonin synthesis [18]. Pernicious lifestyles (caffeine, alcohol, shift work) and some diseases, like type II diabetes, are known to decrease melatonin levels [19,20].

This molecule is associated with a broad spectrum of physiological processes, including regulation of blood pressure, cardiovascular autonomic function, and immune response (Figure 1). Furthermore, it plays key roles as a circadian rhythm regulator, including the modulation of sleep–wake cycles and antioxidant defenses [1,5,14,21]. Melatonin can exert its effect through endocrine, autocrine and paracrine mechanisms. Over the course of evolution, melatonin has acquired additional functions, moonlighting as a signaling molecule involved in the regulation of photoperiodic responses through endocrine pathways [5,15]. Its amphiphilicity allows the molecule to enter any cell, compartment or body fluid [22]. Melatonin mainly exerts its functions through binding to G protein-coupled membrane receptors (Melatonin Receptor 1 (*MT1*), *MT2* and *MT3*), the orphan nuclear receptors (such as orphan receptor retinoid Z receptor beta (*RZR-β*), retinoic acid-related orphan receptor alpha (*ROR-α*)) function is currently controversial, as well as through its interaction with intracellular proteins such as Calmodulin [23]. *MT1* receptors, expressed in the central nervous system and cardiovascular tissues, contribute to the regulation of circadian rhythms and vascular tone [24,25,26]. *MT2* receptors are located especially in the retina and brain and exert functions related to retinal physiology and to the inflammatory response [24,25,26]. They are also involved in circadian rhythm and cardiac vessel dilation (Figure 1). Finally, the *MT3*, renamed from Mel2 receptors, plays a similar role to *MT2* [23,27].

## 2. Antioxidant Function of Melatonin

Melatonin’s initial and most conserved function across diverse organisms is its role as an antioxidant and free radical scavenger, which is vital for cellular homeostasis [5,15]. Melatonin acts as a unique antioxidant, neutralizing more than one molecule, moving across physiological barriers, stimulating endogenous antioxidant enzymes, improving the efficacy of other antioxidants and protecting mitochondria. Thus, melatonin is considered significantly more effective than classical antioxidants [5,28]. Melatonin’s antioxidant functions include (Figure 1):(a)direct free radical scavenging [29,30,31],(b)stimulation of antioxidative enzymes [32,33,34],(c)increasing the efficiency of mitochondrial oxidative phosphorylation [10,35,36].

These functions are mediated by melatonin, but also by the three main metabolites of melatonin: cyclic 3-hydroxymelatonin (3OHM), N-acetyl-N-formyl-5-methoxyquinuramine (AFMK) and N-acetyl-5-methoxyquinuramine (AMK) [37]. AFMK can be formed through enzymatic, pseudoenzymatic, and nonenzymatic metabolic pathways [37,38]. AMK is also generated through deformylation of AFMK [37].

### 2.1. Free Radical Scavenging

The use of oxygen represented an important cost to aerobic organisms. Oxygen metabolism leads to the generation of free radicals, including reactive oxygen species (ROS) and reactive nitrogen species (RNS). Their accumulation disrupts cellular homeostasis and is detrimental to cell survival [39]. Both melatonin and its metabolites can initiate a cascade to neutralize free radicals such as superoxide anions (O_2_^−^), hydroxyl radicals (·OH), nitric oxide (·NO) and peroxynitrite anions (ONOO^−^) [40,41]. A study carried out by Pieri et al. (1996) showed that melatonin exhibits antioxidant activity approximately three times greater than reduced glutathione (GSH) and about two times greater than vitamins C and E [21]. Furthermore, melatonin’s metabolite AMK is a potent scavenger of singlet oxygen (^1^O_2_) and NO [39], whereas the metabolite 3OHM effectively scavenges ·OH, 2, 2′-azino-bis(3-ethylbenzthiazoline-6-sulfonic acid) (ABST), and peroxide radicals [37,38].

### 2.2. Stimulation of Antioxidative Enzymes

In addition to its free radical scavenging capacity, melatonin activates class III deacetylase sirtuin family members. Melatonin promotes SIRT3-mediated deacetylation of NF-κB, thereby reducing the inflammatory response, and also induces the deacetylation of SOD2, increasing its enzymatic activity [42]. SIRT3 deacetylates FoxO3a, thereby activating an antioxidant response by modulating superoxide dismutase (SOD) and catalase (CAT) enzymes [32,43]. These SIRT3-dependent effects are mediated by the *MT2* receptor [44]. Melatonin also upregulates the expression of antioxidant enzyme genes through *MT1/MT2* receptors, including *glutathione peroxidase* (*GPx*) [21,32,33,43]. This effect is mediated by a rise in γ-glutamylcysteine synthase activity, which is the rate-limiting enzyme of the GSH synthesis pathway [14,33,34]. As an indirect antioxidant it can also downregulate prooxidative and pro-inflammatory enzymes, such as *cyclooxygenase 2* and inducible *nitric oxide synthase (iNOS)* [39] (Figure 1).

### 2.3. Efficiency of Mitochondrial Oxidative Phosphorylation

Melatonin may be synthesized in the mitochondria or elsewhere. But its presence in this organelle can be further increased by entering through the oligopeptide transporters 1 (PEPT-1) and PEPT2. The presence of *MT1* and *MT2* melatonin receptors on the mitochondrial membrane has been described in several reports [45,46]. These indicate the relevance of melatonin neutralizing ROS produced naturally by the electron respiratory chain complexes [35]. Studies by Okatani et al. (2003) showed that melatonin restored the respiratory physiology of aged mouse hepatocytes, specifically at the level of complexes I and IV [47]. Moreover, melatonin has been shown to stimulate oxidative phosphorylation and promote ATP production in neuronal and hepatocyte mitochondria [35]. In addition, melatonin increases the activity of uncoupling proteins (UCP) by direct action or by upregulating their gene expression [36]. These proteins transport intermembrane protons back to the mitochondrial matrix, reducing the membrane potential (Δψ), thus accelerating electron transport through the electron transport chain. This phenomenon drastically reduces electron leakage and, consequently, ROS formation [36]. Another protective function of melatonin through the mitochondria is related to apoptosis. Through its antioxidant activity, melatonin reduces mitochondrial damage, thereby preventing the release of pro-apoptotic factors, such as cytochrome c, and inhibiting the intrinsic apoptotic pathway [48]. It has also been reported that when melatonin binds to the mitochondrial *MT1* receptor, it activates Gαi, which blocks adenylate cyclase activity and inhibits stress-induced cytochrome c release, thereby dampening caspase activation [10]. This protective role is context-dependent, as melatonin can also have pro-apoptotic role in a cancer scenario (we will explore it in the next sections).

Beyond its classical antioxidant role, melatonin’s ability to preserve mitochondrial integrity places this molecule at a strategic crossroads between cellular metabolism and immune regulation. Because immune cell activation, differentiation, and effector functions are tightly dependent on redox balance and mitochondrial fitness, melatonin-mediated control of oxidative stress has direct consequences for both innate and adaptive immune responses. These interactions are explored in the following section.

## 3. Immunomodulatory and Anti-Inflammatory Actions of Melatonin

The first line of defense (after the skin) against pathogens and diseases is the immune system. Redox homeostasis and mitochondrial function critically shape immune cell fate, cytokine production, and inflammatory outcomes. In this context, melatonin emerges as a key immunometabolic regulator, coordinating antioxidant defenses with immune activation or suppression depending on the scenario. The immune system is a complex biological network composed of different cell types, tissues, and organs that releases specific proteins and molecules and provides protection [49,50,51]. There are two main types of immunity: adaptative, or specific response; and the innate, or non-specific response [49,50,51].

The adaptive immune response is more specific, and its magnitude increases with successive exposures to a specific insult. This type of response is slower since it requires cellular activation [49,50,51]. The cell types involved in this type of response are T and B lymphocytes, along with various cytokines and antibodies (humoral immunity) [49,50,51].

The innate response includes defense mechanisms that are present before infection occurs, enabling a rapid response. The main cellular components include macrophages, neutrophils, basophils, eosinophils, and natural killer (NK) cells. These cells release various soluble factors, such as cytokines: tumor necrosis factor alpha (TNF-α), (IL)-1β, IL-6 and IL-8.

Interleukins and cytokines produced in the immune response modulate the synthesis and release of melatonin. For instance, an increase in the inflammatory cytokine IL-1β stimulates the signals of the endocrine system (melatonin and cortisol) to counteract the secretion of IL-1β [52]. This reveals an intimate relationship between the immune and the endocrine system. Many immune cells, such as lymphoid cells, express melatonin receptors and a bidirectional communication circuit among both systems (immune and melatonin) is established [52,53]. As an example of the role of melatonin in the immune system, Maestroni et al. (1986) demonstrated in mice that inhibition of melatonin synthesis reduced the adaptive, cellular, and humoral response, which could be reversed by melatonin administration [54]. In addition, circulating leukocytes and monocytes are a well-established source of extra pineal melatonin [52,55].

### 3.1. Adaptative Immune Response

#### 3.1.1. T Cells

T cells develop from bone marrow (BM)-derived thymocyte progenitors and mature in the thymus. Broadly speaking, this cell population is composed of CD4+ and CD8+ T lymphocytes [51,56]. They become activated and undergo clonal expansion following interaction of the T cell receptor (TCR) with processed antigen presented on major histocompatibility complex (MHC) molecules of antigen-presenting cells (APCs) (signal 1) [51,56]. In addition, they can interact with co-stimulatory molecules (signal 2) and cytokines (signal 3) [51,56]. Upon activation, these cells release cytokines which trigger cell death in the target cell, mainly through direct interaction [51,56]. Depending on the cytokine-mediated activation signal, these T cells can be activated toward different programs [57]. IFN-γ and IL-12 drive the differentiation of Th1 cells responsible for protection against intracellular viruses and bacteria. IL-13 and IL-4 cytokines induce differentiation towards a Th2 phenotype, acting against helminth infections [57]. This phenotype also facilitates tissue repair and can contribute to chronic inflammation, typically in allergies and asthma. In contrast, IL-6, IL-21 and IL-23 stabilize the Th17 lineage, characterized by its response against extracellular pathogens [57]. The Th9 program, activated by the cytokines IL-4 and TGF-β, plays an essential role in infectious diseases, cancer, and autoimmune diseases [57]. There is another subpopulation of CD4+ T lymphocytes called T_Reg_, induced by TGF-β, which seem very relevant for damage resolution. T_Reg_ are focused on suppressing immune system cells and preventing exacerbated reactions [56].

In murine T cells, melatonin exerts distinct effects depending on the immunological context. Exogenous melatonin promotes both the activation of T cells and their proliferation [53,58]. This effect occurs through a reduction in IL-10, and the increased level of the T cell activation marker CD69 [53,58]. Additionally, there is a notable increase in the abundance of proliferation markers such as Ki67 and Bcl2 [58]. The underlying mechanism may involve *MT1* receptors, as *MT1* antagonists limit T cell proliferation [58]. Melatonin might not only control proliferation. Stimulation of ovalbumin-activated T cells with melatonin increased IL-4 levels, contributing to the clonal expansion of Th2 cells in vivo. However, melatonin did not increase the production of this interleukin in naïve T cells not activated with ovalbumin, suggesting that the effect of melatonin only occurs in antigen-activated T cells [59]. In a model of AIDS infection, melatonin administration was associated with an increase in IL-2 and IFN-γ levels and a decrease in Th2 cytokine production. This was linked to an increase in the proliferation of T-cell as well as an increase in Th1 cytokine secretion, having a protective effect against retrovirus infection [60].

Following cell activation, most T cells undergo apoptosis. But a small population persists in the form of memory T cells (T_M_) [61]. These T_M_ can differentiate into central memory T cells (T_CM_), effector memory T cells (T_EM_), memory stem T cells (T_SCM_), resident memory T cells (T_RM_), and follicular memory T cells (T_FH_) [61]. This classification is based on cell surface expression of homing and selectin molecules, production of effector cytokines, and location [61]. Studies conducted by Álvarez-Sánchez et al. (2015) showed that melatonin increased the population of T_CM_ cells in the central nervous system in an Experimental Autoimmune Encephalomyelitis (EAE) model [62]. These T cells are the most frequent population of immune cells in the cerebrospinal fluid of healthy humans [62]. Melatonin also decreased the population of T_EM_ cells, which are more characteristic in EAE patients [62]. These studies showed that melatonin reduces the level of CD44, a marker of effector cells, and contributes to the increase in T_Reg_ cells, which diminishes the immune response, alleviating the symptoms of the disease [62]. In myasthenia gravis, an autoimmune disorder, melatonin decreases the Th1 and Th17 response, as well as the production of pro-inflammatory cytokines, which have benefit effects due to the basis of the disease [63]. Taken together, these results suggest that melatonin has a dual environment-dependent role. It acts as a stimulator of the immune response under conditions of immunosuppression, for instance, in a model of AIDS infection [60]. But melatonin acts as a negative modulator during exacerbated immune responses, like in myasthenia gravis [63] or an EAE model [62].

#### 3.1.2. B Cells

B cells recognize and respond specifically to each antigenic epitope without the need to interact with APCs [64]. They originate from hematopoietic stem cells in the bone marrow [64]. They mature within secondary lymphoid organs and differentiate into plasmacytes after activation by direct interaction between antigen and B-cell receptor (BCR) [64]. These cells also present co-receptors and co-stimulatory molecules, like T lymphocytes, which can be regulated by the BCR [64]. Plasma cells, or plasmacytes, have the capacity to synthesize antibodies (AB). There are different populations of B-cells. Memory B cells, which are long-lived and have two subpopulations depending on the origin and expression of markers. Then, regulatory B cells (B_Reg_), which restrict excessive inflammatory responses. These B_Reg_ cells ensure tolerance through the release of cytokines and interleukins, such as IL-10, that occur in all inflammatory reactions. This release of cytokines is exacerbated during autoimmune diseases or unresolved infections [65].

Only a limited number of studies have investigated the biological effects of melatonin on B lymphocytes [27]. Melatonin stimulation of calmodulin-PLA2-5-lipoxygenase signaling pathway fosters a pro-inflammatory response in leucocytes. Melatonin specifically binds to calmodulin, releasing Calcium, which activates the lipoxygenase [66]. This encodes for a key enzyme in the biosynthesis of leukotrienes, which are involved in inflammatory and allergic processes. This may suggest a melatonin role in these immunological reactions through B lymphocytes [27]. In vivo studies in chickens have shown that exogenous administration of melatonin stimulates the proliferation of B lymphocytes [27]. Melatonin also promotes B cell proliferation in mouse models [27]. It has been suggested that melatonin may be a checkpoint regulator in early B cell development, as it prevented apoptosis of bone marrow B cells during early differentiation when orally administered to mice [27,67]. Studies conducted by Luo et al. (2020) suggested that the role of melatonin in B cell activation depends on concentration and timing of the administration [68]. For instance, melatonin treatment for two weeks inhibited the expression of stress response genes, like *p38* and *NF-κB*. The same treatment for four weeks caused the opposite effect, recovering *p38* and *NF-κB* levels to a basal state. Similar results were found when different doses were used [52].

### 3.2. Innate Immune Response

In addition to its well-documented effects on adaptive immunity, melatonin has also been shown to modulate cells of the innate immune system. For instance, in vivo experiments by Liang et al. (2024) demonstrated that treatment of aged mice with melatonin for 1–2 weeks enhanced the proliferation of innate immune cells in the spleen, liver, and bone marrow [69].

**Natural killer (NK)** cells are innate immune lymphocytes whose activity is regulated by a balance of inhibitory and activating receptors [70]. Unlike T cells, NK cells do not require antigen presentation by APCs because they can recognize their target cells directly [70]. They discriminate between healthy and abnormal cells through recognition of MHC class I molecules, which engage inhibitory receptors [71]. Upon activation, NK cells secrete a variety of cytokines, such as interferon-γ (IFN-γ), as well as growth factors including granulocyte–macrophage colony-stimulating factor (GM-CSF) and diverse chemokines [70]. NK cells kill target cells by releasing perforin and granzymes or through contact-dependent mechanisms mediated by Fas–FasL and TRAIL, collectively known as death-ligand pathways. In addition, NK cells mediate antibody-dependent cellular cytotoxicity (ADCC) [70,71]. Melatonin promotes NK cell degranulation and IFN-γ secretion via activation of the JAK3/STAT5 and T-bet signaling pathways [69].

**Mast cells** are hematopoietic-derived immune cells. Under normal conditions, mature mast cells do not circulate in the bloodstream. Instead, mast cell progenitors migrate into peripheral tissues, where they differentiate under the influence of stem cell factors and various cytokines [72]. These cells contain large cytoplasmic granules, like those in neutrophils, which store inflammatory mediators such as histamine and heparin [72]. Their primary mechanism of action involves the binding of immunoglobulin E (IgE) during allergic reactions. They can also recognize harmful antigens and release inflammatory mediators that contribute to the elimination of pathogen [72]. Evidence from Caiyun Huo et al. (2023) highlights the role of melatonin in innate immunity, showing that mice with aberrant melatonin production exhibited increased mortality following H1N1 viral infection [73]. This heightened mortality was attributed to melatonin-mediated suppression of mast cell activation, which in turn reduced the inflammatory response through regulation of cytokine secretion [73].

**Dendritic cells (DCs)** are among the most professional antigen-presenting cell types [74]. In the conventional presentation pathway, exogenous antigens are displayed on MHC class II molecules to CD4^+^ T cells, whereas endogenous antigens are presented on MHC class I molecules to CD8^+^ T cells [74]. DCs are also highly efficient at presenting exogenous antigens on MHC class I molecules through an unconventional pathway known as cross-presentation (Liu, 2015) [74]. Huang et al. (2024) reported that melatonin influences T cell differentiation by suppressing NF-κB activation in DCs, thereby indirectly shaping adaptive responses through modulation of innate immune cell activity [75].

**Neutrophils** are the most abundant leukocyte in humans and are essential components of the innate immune response. They can rapidly kill invading pathogens by phagocytosis [76]. Although these cells are not classic cytokine producers and do not function as APCs, they play a central role in inflammation and release a variety of granule proteins, such as calreticulin [76]. More recently, Tai et al. (2025) described an additional neuroimmune facet of melatonin’s action. In a murine model of transient focal cerebral ischemia, melatonin administration reduced neutrophil infiltration and attenuated microglial and macrophage activation in the brain [77].

**Macrophages** represent another type of professional antigen-presenting cell, and their principal function is phagocytosis [78]. Macrophages exhibit binary M1/M2 polarization states in response to specific stimuli, playing a central role in immune regulation [78]. Classically activated M1 macrophages, induced by stimuli such as lipopolysaccharide (LPS) and IFN-γ, adopt a predominantly aerobic glycolytic metabolism and display a tumor-suppressive profile characterized by the release of IL-6, IL-12, TNF and iNOS [79,80]. Alternatively activated macrophages (M2) promote the resolution of inflammation, healing, and phagocytosis. Their metabolism depends on oxidative phosphorylation, and they show the activation of autophagy. This latter phenotype has a pro-tumoral profile. Membrane melatonin receptors (*MT1*/*MT2*) were identified on murine peritoneal macrophages as early as 1999, and pharmacological blockade with the non-selective *MT1*/*MT2* antagonist luzindole diminished phagocytic activity in RAW264.7 macrophages [81,82]. Melatonin modulates macrophage polarization in an environment-dependent manner via pathways such as NF-κB [83]. It negatively regulates *iNOS* mRNA expression through NF-κB inhibition, thereby reducing ROS generation that can impair electron-transport chain function [82]. Yet the effects are context-specific: classic studies reported that human monocytes exposed to LPS and then treated with melatonin differentiated toward a pro-inflammatory, cytotoxic macrophage phenotype, whereas monocytes pre-activated with melatonin and subsequently challenged with LPS exhibited pro-oxidant and pro-inflammatory profiles [84]. In RAW264.7 cells, danger signals that activate NF-κB have been shown to induce melatonin synthesis, a mechanism involved in both the initiation and resolution of the innate immune response [85]. In addition, Wang et al. (2023) reported that 0.3 mM melatonin induced apoptosis in RAW264.7 cells through Bmal1 and the MAPK-p38 pathway [86]. This effect was associated with upregulation of Bmal1, reduced levels of phosphorylated MAPK-p38, and a subsequent increase in ROS [86].

## 4. Melatonin in Cancer: Mechanistic Insights and Therapeutic Potential

Chronic inflammation, immune dysregulation, and metabolic stress are now recognized as central drivers of tumor initiation and progression. Given melatonin’s dual capacity to modulate oxidative stress and immune function, its relevance extends naturally to cancer biology, where redox imbalance, mitochondrial dysfunction, and immune evasion converge. Cancer remains one of the leading causes of mortality worldwide [87]. Cancer is a multifactorial disease, a collection of disorders arising primarily from the body’s own cells, which acquire abnormal growth potential and the capacity to invade and metastasize beyond their tissue of origin [88]. These changes are driven by genetic alterations in key cellular pathways, such as DNA repair, proliferation control, and defense mechanisms [88]. The hallmark of cancer is cellular immortalization, wherein tumor cells acquire the ability to proliferate independently of external growth signals while evading apoptosis [89,90,91]. Tumor cells also undergo profound metabolic reprogramming through the Warburg effect, shifting energy production from mitochondrial oxidative phosphorylation toward aerobic glycolysis in the cytosol [92].

Beyond genetic alterations, tumor progression is critically shaped by the spatial organization and metabolic heterogeneity of the tumor microenvironment (TME). Solid tumors display marked gradients of oxygen, nutrients, and metabolites, which profoundly influence immune cell infiltration, macrophage polarization, and immune evasion mechanisms. Hypoxic tumor regions favor stabilization of HIF-1α, driving metabolic reprogramming toward aerobic glycolysis and lactate accumulation, a hallmark of aggressive tumor behavior. This metabolic remodeling not only supports cancer cell survival but also actively suppresses antitumor immunity. Tumor-associated macrophages (TAMs) constitute a dominant immune population within the TME and exhibit remarkable functional plasticity. Depending on local cues, TAMs adopt pro-inflammatory, tumor-suppressive M1-like phenotypes or anti-inflammatory, tumor-promoting M2-like states. Hypoxia, tumor-derived metabolites, and immunosuppressive cytokines skew macrophage polarization toward M2 phenotypes, which support angiogenesis, extracellular matrix remodeling, and immune suppression. Importantly, melatonin has emerged as a potent modulator of macrophage metabolism and polarization, favoring antitumoral M1 programs while restraining M2-associated functions [65].

Metabolic reprogramming and immune escape are tightly interconnected processes in cancer. Elevated glycolysis and lactate production impair cytotoxic T lymphocyte and NK cell function, promote T cell exhaustion, and enhance immune checkpoint expression, including PD-L1. By targeting mitochondrial metabolism, through the inhibition of HIF-1α–dependent pathways, and restoring oxidative phosphorylation, melatonin disrupts these immune evasion circuits. Thus, melatonin acts not only as a direct antitumoral agent; it reconfigures the TME, linking metabolic control to immune surveillance [82].

Due to its anti-inflammatory role and its ability to modulate the immune system, melatonin has been proposed as an antitumoral agent, targeting key events of cancer initiation, progression, and metastasis. Experimental evidence highlights the multifaceted anticancer actions of melatonin, ranging from direct modulation of tumor cell survival to TEM regulation and metabolic reprogramming. Laborda-Illanes et al. (2023) demonstrated that vascular endothelial growth factor (VEGF) and MCF-7 breast cancer cells induce angiogenic sprouting in three-dimensional spheroids of human umbilical vein endothelial cells (HUVECs) [93]. In this setting, 1 mM melatonin effectively suppressed neovascular outgrowth by inhibiting angiogenesis (Figure 2). Complementary studies by Zeppa et al. (2024) revealed that melatonin reduced the viability of pancreatic ductal adenocarcinoma cell lines PANC-1 and MiaPaCa-2 through both apoptotic and necroptotic mechanisms [94] (Figure 2). Combined treatment with antitumor agents (cannabidiol or sorafenib) and melatonin inhibited proliferation and enhanced indicators of cell death [94]. Melatonin metabolites also display anticancer potential, as they inhibited proliferation in SKMEL-188 human melanoma cells without affecting melanogenesis [95]. Pinealectomy studies highlight melatonin’s protective role. The removal of the pineal gland in rats led to larger and thicker oral squamous cell carcinomas, potentially due to loss of melatonin’s capacity to inactivate ROS-dependent Akt signaling and reduced *cyclin D1*, *PCNA*, and *Bcl-2* expression [96].

Immune checkpoint regulation has emerged as a significant anticancer mechanism [97]. Programmed death-ligand 1 (PD-L1) is key for immune evasion. Upon PD-1 binding to PD-L1, T cell proliferation and activation are suppressed [97]. Melatonin reduces *PD-L1* expression in hepatocellular carcinoma through hypoxia-inducible factor-1α (HIF-1α) and MAPK-dependent pathways [98] (Figure 2). An effect replicated in a model of non-small-cell lung cancer and head-and-neck squamous cell carcinoma (HNSCC) [99]. Melatonin treatment also blocked epithelial-to-mesenchymal transition (EMT) in HNSCC [100] (Figure 2). This effect, tested in vitro and in vivo, was mediated through the inhibition of ERK1/2–FOSL1 signaling, thereby enhancing CD8^+^ T-cell infiltration and potentiating antitumoral immunity [100].

Tumors are often hypoxic and exhibit elevated HIF-1α levels. This promotes expression of *pyruvate dehydrogenase kinase 1* (*PDK1*), which inhibits the pyruvate dehydrogenase complex (PDC) and thereby prevents pyruvate conversion to acetyl-CoA, fostering a Warburg-type glycolytic phenotype. Acetyl-CoA is also a co-factor for the rate-limiting enzyme in melatonin synthesis, the aralkylamine N-acetyltransferase (AANAT). Thus, this metabolic switch may limit mitochondrial synthesis of melatonin [101,102]. When melatonin is used to treat cancer, for instance, in ovarian cancer, melatonin counteracts this metabolic switch by inhibiting PDK activity, restoring pyruvate flux into acetyl-CoA, promoting mitochondrial oxidative metabolism, and reversing the Warburg effect [102] (Figure 2).

Clinical applications have been explored using melatonin as an adjuvant with traditional treatments [103]. Oral melatonin was administered along with the radiofrequency ablation (RFA) in early-stage non-small-cell lung cancer patients with multiple ground-glass nodules [103]. An oral dose of 5 mg/day melatonin for 12 months, 1 week after RFA treatment, was prescribed to patients, without any indication of the time of day, although the timing would impact the efficacy of the treatment [103]. This therapy yielded significantly fewer or smaller untreated nodules compared with conventional surgery [103]. The beneficial effects were mediated by a combined induction and RFA of the NK-cell antitumor responses [103]. Melatonin, as in the previously described work [102], reprogrammed residual tumor metabolism toward oxidative phosphorylation. HIF-1α and other important transcriptional factors associated with malignancy were downregulated, while oxidative metabolism was stimulated. Thus, melatonin reduced hypoxia in the tumor microenvironment, consistent with Warburg effect reversal [103]. Melatonin has also been investigated for mitigating cancer-related fatigue (CRF), a common adverse effect of conventional therapies [104]. In patients with invasive breast carcinoma receiving chemotherapy and radiotherapy, 18 mg/day oral melatonin reduced CRF and improved quality of life [104]. However, other randomized controlled trials in early-stage breast cancer or ductal carcinoma in situ reported no significant benefit [105]. In both studies, the dose was very similar, but the time of administration was not indicated, which can have an enormous impact on the efficacy of the treatment [106,107].

## 5. Modulation of the Tumor Immune Microenvironment

Tumor progression cannot be fully understood by focusing exclusively on malignant cells. Cancer emerges from dynamic interactions between tumor cells and their surrounding microenvironment. This TME represents the functional interface where melatonin’s antioxidant, immunomodulatory, and metabolic effects converge. The immune response to tumoral cells includes three major phases [108,109]:Elimination. Innate and adaptive immune components act to eradicate emerging tumor cells.Equilibrium. Tumor cells with low immunogenicity survive immune pressure and continue unchecked proliferation.Escape. Tumor cells downregulate MHC-I expression, impairing immune recognition and culminating in the emergence of clinically detectable tumors.

Tumor cells can evade immune surveillance by increasing the production of proteins that suppress the immune system [89,90,91]. Treatment challenges arise from both types of immune evasion (2 and 3) and the cellular heterogeneity of the TME [110,111]. The term TME was first introduced in the 19th century by Paget, who proposed that bidirectional communication between tumors and their surrounding milieu was critical to understanding relapses, metastasis, and drug resistance [112,113]. The TME contains non-malignant cells, including immune cells, cancer-associated fibroblasts (CAF), endothelial cells (EC), pericytes, and tissue-specific cell types such as adipocytes or neurons, whose composition varies depending on tumor location [110,111,113]. Once considered passive bystanders, TME cells and their secreted factors are now recognized as active participants in tumorigenesis and as potential therapeutic targets [111,113,114]. The immune cells infiltrated within tumors can be broadly categorized into tumor-antagonizing populations: effector T lymphocytes (CD8^+^ cytotoxic and CD4^+^ helper subsets), NK cells, DCs, and M1 macrophages; and tumor-promoting populations: T_Reg_, TAMs and myeloid-derived suppressor cells (MDSCs) [110]. Tumor cells actively shape the TME by recruiting and reprogramming host cells, remodeling the vasculature, and reorganizing the extracellular matrix (ECM). This process depends on complex heterotypic interactions between malignant and non-malignant components [111].

Within the TME, adaptive immune cells (principally T and B lymphocytes) play pivotal roles in tumor control and immune evasion. A key mechanism is T cell “exhaustion”. This term refers to the diminished cytokine production and increased levels of inhibitory receptors of lymphocytes, which impair their activation [115]. In solid tumors, CD8^+^ T lymphocytes mediate direct tumor cell elimination, whereas T_Reg_ represent the predominant immunosuppressive CD4^+^ subset [114]. The presence of effector CD4^+^ T cell subsets in both circulation and the TME has been linked to favorable prognosis across multiple cancer types [116,117]. CD4^+^ T cells exert dual and context-dependent effects: **Th1 cells** promote tumor senescence and drive tumor *MHC-II* expression (anti-tumoral), but they may also facilitate tumor maintenance via pro-angiogenic activity (pro-tumoral); **Th2 cells** can recruit cytotoxic eosinophils (anti-tumoral) but also favor the accumulation of immunosuppressive M2 macrophages (pro-tumoral) [116,117]. Melatonin increases Th1 cells in mice with tumors [118] and reduced the proliferation of T_Reg_, which have a pro-tumoral role [119]. Metabolic constraints within the TME further compromise T cell function [120]. Elevated lactate concentrations and extracellular acidity inhibit cytokine production by both CD8^+^ and CD4^+^ T cells, markedly reducing their cytotoxic potential [120]. Melatonin inhibition of glycolysis may decrease extracellular acidification and thereby support T cell function [121]. B lymphocytes contribute to tumor control primarily through the production of antigen-specific antibodies, which promote antibody-dependent cellular cytotoxicity (ADCC) by NK cells [122]. In addition, direct antigen recognition via the B cell receptor (BCR) can trigger the release of proteolytic enzymes such as granzyme B (GzmB), inducing apoptosis in tumor cells [122,123].

Among innate immune effectors, **NK cells** recognize tumor targets independently of *MHC-I* expression, complementing CD8^+^ T cell surveillance [124]. NK-mediated cytotoxicity involves perforin-dependent pore formation and granzyme-induced apoptosis, as well as death receptor engagement via FasL and TRAIL [125]. They also secrete IFN-γ and TNF-α, which can suppress tumor proliferation and survival. Nevertheless, NK cells often exhibit functional impairment within the TME [114]. Tumor cells modulate ligands for NK cell receptors, which activate the Natural Killer group 2, member D (NKG2D) and NKp30. In addition, tumor cells can inhibit killer immunoglobulin-like receptors (KIR) [124,126]. It is described that melatonin amplifies lytic activity of NK cells [119]. Also, melatonin induces the release of cytokines, such as Il-6, Il-12 and IFN-γ, which increase NK cell proliferation [119]. This activation and proliferation will boost the immune response in TME by reducing its volume and immunosuppressive environment. **Neutrophils**, including tumor-associated neutrophils (TANs), exhibit polarized phenotypes: antitumorigenic N1 and pro-tumoral N2 subsets [114]. Their canonical antimicrobial role, phagocytosis, and ROS generation could exert anti-tumoral functions, but ROS in the TME induces mutagenesis and supports tumor progression [127]. Karadas et al. in 2021 indicate that melatonin reduced the number of neutrophils in peripheral blood and decreased their infiltration into the TME in mouse model of breast cancer [128]. **Myeloid-derived suppressor cells** (MDSCs) expand in cancer patients, further suppressing T cell activity, enhancing *PD-L1* expression, secreting ROS and RNS (e.g., nitric oxide), downregulating genes required for T cell extravasation and tumor infiltration, and promoting angiogenesis and pre-metastatic niche formation [129]. Melatonin reduces the accumulation of MDSCs by decreasing TGF-β secretion by tumor cells [130]. TAMs, **Macrophages** that infiltrate TME, exemplify the functional plasticity of myeloid cells [131]. However, due to macrophages’ relevance in the TME a whole section about the melatonin actions in the TAMs is next to this one.

In addition to these hematopoietic components, non-hematopoietic stromal cells such as **CAFs** orchestrate multiple tumor-supportive mechanisms [132]. CAFs remodel the extracellular matrix, secrete pro-angiogenic factors (VEGFA, PDGFC, CXCL12, osteopontin), and recruit endothelial and myeloid cells, thereby facilitating neovascularization [132]. They also produce TGF-β and CXCL12, fostering T_Reg_ and M2 macrophage differentiation while impairing T cell recruitment and activation. Melatonin prevents the conversion of fibroblasts into CAFs by reducing the production of metalloproteinases 2 and 9 in gastric cancer cells [133]. This helps to disrupt cancer progression.

The immune system exerts a determinant influence on cancer evolution. The modulation of the immune cells through different approaches, including pharmacological and genetic ones, has been proposed as a therapeutic target for cancer [1,118]. Here melatonin is a potential adjuvant for cancer prevention and therapy [134].

## 6. Macrophages as a Therapeutic Target of Melatonin in Cancer

Macrophages, particularly TAMs, account for half of the cells of the TEM and are key for both tumor progression and regression [129]. TAMs show different metabolic and functional states in response to microenvironmental cues, from pro-inflammatory, classically activated M1-like programs to anti-inflammatory, alternatively activated M2-like programs [129]. M1 effector mechanisms include ROS and RNS production with direct, albeit relatively slow, cytotoxicity toward tumor cells, and rapid antibody-dependent cellular cytotoxicity (ADCC). M1 activity is also associated with increased numbers of activated NK cells with death-receptor ligands, thereby amplifying tumor cell killing [81,82]. M2-like TAMs support tumor progression through well-documented mechanisms [82]. In different malignancies, the prevalence of M2-polarized TAMs correlates with reduced patient survival, increased metastatic dissemination, and enhanced tumor proliferation [135,136,137]. M2 phenotypes are associated with expression of proangiogenic factors (VEGF, basic fibroblast growth factor (bFGF) and others), secretion of matrix-remodeling enzymes (matrix metalloproteinases, serine proteases, cathepsins) that degrade collagens and extracellular-matrix components, promotion of endothelial proliferation and neovascularization, and facilitation of stromal and tumor cell migration [82]. M2 macrophages can also suppress CD8^+^ T cell proliferation via L-arginine metabolism [82]. Tumor-derived metabolites further modulate phenotype towards M2. For example, extracellular lactate produced by tumor metabolism promotes an M2-like activation through histone lactylation and related metabolic–epigenetic crosstalk [138,139].

Melatonin has been reported to limit or prevent M2 polarization [140]. Given the central role of the immune system in cancer control and the capacity of melatonin to enhance immune function, the modulation of macrophage phenotype by melatonin has emerged as a promising therapeutic avenue [131]. This could be mediated by multiple mechanisms: suppression of senescence, enhancement of antigen presentation, interruption of tumor-promoting inflammatory circuits, metabolic reprogramming, and modulation of immune checkpoints [131].

Selective clearance of senescent macrophages could, in principle, reverse an immunosuppressive TME. This would create an immunostimulatory TME and would disrupt tumoral vascular networks. Thereby reducing nutrient and oxygen supply to cancer cells [135]. Recent mechanistic work by Zhang et al. (2025) demonstrated in an experimental breast cancer model that, compared with melatonin-treated conditions, macrophages without melatonin exposure secreted high levels of senescence-associated secretory phenotype (SASP) factors, driving inflammation, macrophage senescence, and immune dysfunction, which contribute to chemotherapy resistance [141]. Melatonin counteracts these effects by upregulating *Trim26*, which in turn blocks activation of the cyclic GMP–AMP synthase–stimulator of interferon genes (*cGAS–STING*) pathway, thereby attenuating macrophage senescence, promoting protumoral response [141] (Figure 3). However, chronic activation of cGAS–STING is known to sustain pro-tumorigenic inflammation and foster an immunosuppressive TME [142]. Melatonin also enhances macrophage-mediated antigen presentation by upregulating *MHC-II* expression, amplifying both specific and non-specific T-cell responses [143] (Figure 3). Melatonin treatment interrupts tumor invasion and migration by reducing MIF (macrophage migration inhibitory factor) and IL-1β levels in THP-1 cells, thereby limiting the crosstalk between TAMs and cancer cells, reducing the inflammation in oral squamous cell carcinoma [144] (Figure 3).

Metabolic reprogramming of TAMs is another relevant therapy [145]. In colostrum-derived macrophages co-cultured with breast cancer cell lines, exposure to melatonin increased the M1/M2 ratio, altering the release of cytokines, particularly TNF-α and IL-8. Melatonin reduced both pro-tumoral and immunosuppressive phenotypes [146]. In a lymphoma model, melatonin inhibited arginase activity, leading to increased nitric oxide (NO) production and decreased urea levels within the TME, culminating in tumor cell death [145] (Figure 3). In relation to this, melatonin alters intercellular communication by modulating cancer cell–derived exosomes [147,148]. In gastric cancer, melatonin-treated macrophages displayed reduced PD-L1 levels, enhancing their antitumor activity. This effect extended to tumor cells themselves and was associated with altered exosomal microRNA content capable of suppressing *PD-L1* expression [147,148] (Figure 3). In lung cancer, melatonin combined with LPS reduced M2 macrophage polarization and induced M2 apoptosis via activation of JAK–STAT, Toll-like receptor, and chemokine signaling pathways [149]. Prolonged LPS exposure, however, induces tolerance and favors M2 polarization, underscoring the importance of temporal dynamics in therapeutic strategies [149]. Inhibition of the JAK2/STAT3 pathway in M2 macrophages diminished PD-L1 production, further relieving T-cell suppression [149]. Thus, melatonin can bias macrophage behavior in divergent directions depending on prior activation state and microenvironmental cues, a property that could be harnessed to reprogram TAMs toward tumor-suppressive phenotypes.

## 7. Conclusions

This review provides an integrative perspective on melatonin that extends beyond its well-established antioxidant properties. This indoleamine acts as a central regulator of immunometabolic processes within the TME. The present work adds conceptual novelty by linking mitochondrial regulation, redox homeostasis, immune modulation, and tumor metabolism within a unified framework. A key advance highlighted here is the role of melatonin as a modulator of immune cell function within the TME, particularly through its capacity to influence TAM polarization and immune evasion mechanisms. By preserving mitochondrial function and reprogramming cellular metabolism, melatonin promotes antitumoral immune responses while counteracting the metabolic and inflammatory conditions that favor tumor progression. This positions melatonin not merely as an adjuvant antioxidant, but as an active modulator of the tumor immune landscape. Despite these promising insights, several unresolved questions remain. The context-dependent effects of melatonin across different tumor types, stages, and immune microenvironments require further clarification. In addition, optimal dosing regimens, chronobiological considerations, and potential synergies between melatonin and immunotherapies or metabolic-targeting agents warrant systematic investigation. Addressing these challenges will be essential to translating the immunometabolic properties of melatonin into effective therapeutic strategies.

Overall, this review underscores the emerging concept of melatonin as a modulator at the intersection of mitochondrial function, immune regulation, and tumor biology, offering a coherent framework to guide future experimental and clinical studies.

## Figures and Tables

**Figure 1 antioxidants-15-00064-f001:**
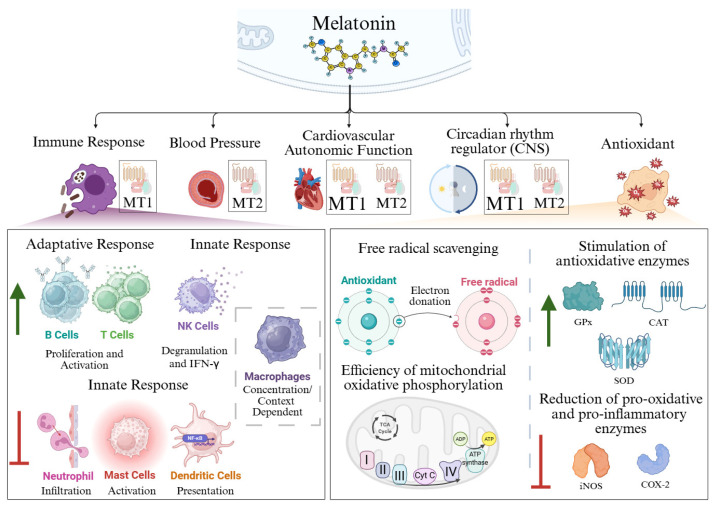
Schematic representation of the biological functions of melatonin. Melatonin exerts multiple physiological functions. Within the immune system, melatonin enhances the proliferation and activation of T and B lymphocytes. In mast cells, melatonin promotes cellular degranulation and IFN-γ release, while also enhancing neutrophil infiltration and antigen presentation by dendritic cells. In macrophages, however, the effects of melatonin appear to be context- and concentration-dependent. Regarding its antioxidant capacity, melatonin functions as a potent free radical scavenger, stimulates antioxidant enzymes, downregulates pro-oxidant and pro-inflammatory enzymes, and improves the efficiency of mitochondrial oxidative phosphorylation. Created in BioRender. Garcia-Macia, M. (18/12/2025) https://BioRender.com/v72fvv9 (accessed on 27 December 2025).

**Figure 2 antioxidants-15-00064-f002:**
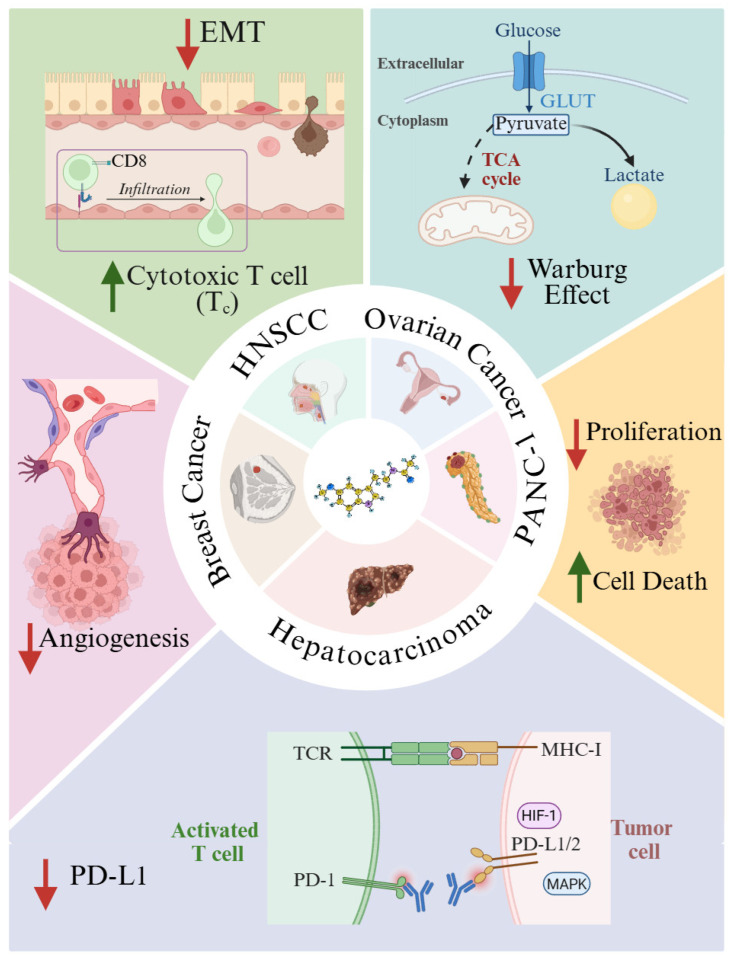
Schematic representation of the antitumor mechanisms of melatonin across different types of cancer. In head and neck squamous cell carcinoma (HNSCC), melatonin enhances the infiltration of cytotoxic CD8^+^ T lymphocytes and suppresses epithelial-to-mesenchymal transition (EMT). In ovarian cancer, melatonin attenuates the Warburg effect by restoring metabolic flux toward acetyl-CoA and enhancing tricarboxylic acid (TCA) cycle activity, thereby improving cellular bioenergetic efficiency. In the pancreatic cancer cell line PANC-1, melatonin increases markers of cell death. In hepatocellular carcinoma, melatonin downregulates PD-L1 expression in tumor cells, preventing T cell inhibition mediated by the PD-1/PD-L1 pathway. In breast cancer, its anti-angiogenic effects restrict nutrient and oxygen supply to the tumor microenvironment, thereby strengthening its overall antineoplastic potential. Created in BioRender. Garcia-Macia, M. (14/11/2024) https://BioRender.com/v72fvv9 (accessed on 27 December 2025).

**Figure 3 antioxidants-15-00064-f003:**
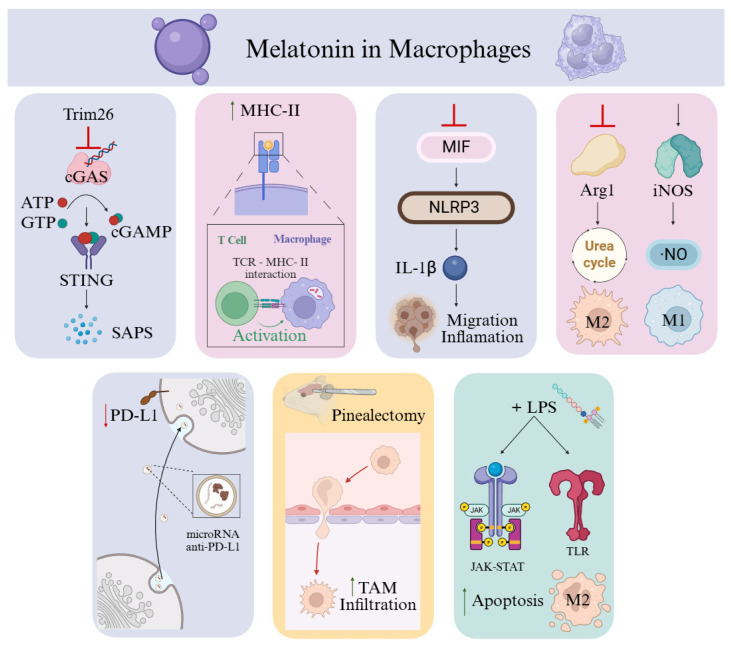
Impact of melatonin on macrophages. Melatonin upregulates Trim26, which contributes to the regulation of macrophage senescence. Melatonin also enhances MHC-II expression, thereby increasing antigen-presenting capacity and amplifying immune activation. Melatonin suppresses the MIF/NLRP3/IL-1β signaling cascade, thereby reducing crosstalk between macrophages and tumor cells. Melatonin inhibits arginase activity, leading to increased nitric oxide (NO) production. Melatonin also modifies the microRNA content of tumor-derived exosomes, leading to decreased PD-L1 expression and attenuated immune suppression. Lack of pineal melatonin in rats increases TAM infiltration and promotes tumor progression. Melatonin combined with lipopolysaccharide (LPS) increases M2 macrophage apoptosis through JAK–STAT and Toll-like receptor (TLR) signaling pathways. Created in BioRender. Garcia-Macia, M. (14/11/2025) https://BioRender.com/v72fvv9 (accessed on 27 December 2025).

## Data Availability

No new data were created or analyzed in this study. Data sharing is not applicable to this article.

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
