# Peer review of "Melatonin at the Crossroads of Oxidative Stress, Immunity, and Cancer Therapy"

_antioxidants, 2026, doi:10.3390/antiox15010064_

Round 1
Reviewer 1 Report
In the manuscript by Lavado-Fernández et al., the authors review the role of melatonin in cancer therapy, with a particular focus on its potential actions as a modulator of the tumour microenvironment (TME). Specifically, the manuscript aims to highlight the role of melatonin in the polarization of macrophages and other adaptive immune cells within the TME of solid tumours.
There are numerous scientific review articles addressing melatonin and cancer, likely numbering in the several hundred when all cancer types and broader onco-endocrinology topics are considered. In addition, the number of publications examining melatonin as an immune-enhancing or immunomodulatory agent in cancer has increased recently and likely exceeds 100. For this reason, it is essential to explain the specific added value of the present review clearly and to emphasize how it advances current knowledge on melatonin and cancer therapy.
There are several aspects that, if addressed, would substantially improve the quality and clarity of the review before publication:
- About the site of melatonin synthesis. The pineal gland primarily produces melatonin. Although several studies suggest that mitochondria may serve as sites of melatonin synthesis, it remains to be determined whether mitochondria constitute the primary site of melatonin production. The text would benefit from a more detailed, up-to-date discussion of this point.
- Mitochondrial melatonin (lines 106–107)
The statement regarding mitochondrial melatonin is controversial. Is melatonin synthesized within mitochondria, or is it synthesized elsewhere and subsequently transported into mitochondria? Please clarify the current evidence and, if applicable, highlight the main hypotheses and limitations. - Melatonin and apoptosis (lines 119–120)
The relationship between melatonin and apoptosis requires clarification. The manuscript suggests that melatonin inhibits apoptosis, but in cancer cells, melatonin has frequently been described as pro-apoptotic. How can these apparently opposite effects be merged? Please discuss the context-dependence of melatonin’s actions (e.g. normal vs tumour cells, dosing, timing). - Disparity in effects (lines 225–226)
It would be beneficial to explain the biological basis of the reported disparity in melatonin’s effects. In addition, are there other hormones or tissue-derived factors that display a similar dual or context-dependent behaviour? Providing such examples would help readers better understand this controversial aspect of melatonin’s activity. - Nuclear receptors and lipoxygenases (line 245). The role of melatonin acting through nuclear receptors was discarded many years ago, and the published article on this point is controversial. Please reconsider the emphasis placed on nuclear receptors and discuss alternative, better-supported mechanisms by which melatonin may modulate lipoxygenase activity.
- Time-dependent effects (line 255). The mechanism underlying the reported time-dependent effect of melatonin should be commented. Is there any mechanistic hypothesis or experimental evidence that could explain this temporal dependence?
- “Novel antitumor agent” (line 344). The sentence stating that “melatonin has been proposed as a novel antitumor agent…” is historically inaccurate. Melatonin was proposed as an antitumor agent as early as the 1960s, and therefore, it cannot be considered “novel” in this regard. Please rephrase this sentence to reflect the historical context more accurately.
- Melatonin synthesis and PKD (line 376). The statement that melatonin synthesis is compromised while, at the same time, it restores PKD is not clear. Please provide more detail on the mechanism involved and discuss the possible interaction between impaired melatonin synthesis and restoration of PKD function.
- Dose, timing, and chronobiology (lines 381–382). How much melatonin was used in the cited studies, and at what time of day was it administered? Melatonin is commonly given in the late afternoon or evening, before bedtime. Does chronobiology (administration time and circadian phase) influence its efficacy as an adjuvant in cancer therapy? A brief discussion of this aspect would considerably enrich the review.
- HIF-1α and hypoxia (line 386). The sentence is difficult to understand as written. It seems to suggest that melatonin reduces HIF-1α activation but not hypoxia in the tumour core. Please rephrase and clarify the mechanism proposed and how melatonin might affect HIF-1α signalling and modify tumour hypoxia.
- Lack of effect in early-stage breast cancer (line 391). The absence of an apparent beneficial effect of melatonin in randomized controlled trials in early-stage breast cancer or ductal carcinoma in situ is an important point. A brief discussion of these negative or neutral results would be highly relevant for this review, as it provides a more balanced and realistic view of melatonin’s translational potential.
- “Reduce the TME” (line 461). The sentence “This will be useful to reduce the TME” is unclear, since “reducing the TME” has no specific biological meaning. Please reformulate to indicate whether you refer to reducing immunosuppression, angiogenesis, stromal content, inflammation, or other particular features of the tumour microenvironment.
- “Melatonin-deficient macrophages” (line 519). The term “melatonin-deficient macrophages” is confusing. Do you mean macrophages from melatonin-deficient mice, or macrophages exposed to melatonin-deficient conditions? Please clarify the experimental model and rephrase accordingly.
- Conclusions
The Conclusions section should be strengthened to summarize the specific interest and novelty of this review clearly. It would be helpful to indicate: (i) what this review adds to the extensive existing literature on melatonin and cancer; (ii) the main conceptual advances regarding melatonin and immune modulation in the TME; and (iii) the key unresolved questions and future directions in the field.
Author Response
Thank you very much for taking the time to review this manuscript. We really appreciate all your comments. They were very insightful, and they improved the quality of the review enormously. We believe that now we can reach a broader audience. Please find detailed responses below, and the corresponding revisions/corrections highlighted/in track changes in the re-submitted files.
Comments 1: There are at least several hundred scientific review articles focused on melatonin and cancer, and likely well over 500 if one includes all cancer types and broad onco-endocrinology reviews. For this reason, it would be beneficial to highlight the specific interest of this review and how it advances knowledge of melatonin and cancer therapy.
Response 1: Thank you for pointing this out. We agree with this comment, and we have modified the abstract (page 1), so it will be much more specific. Our changes are marked in red.
Comments 2: Again, the conclusions need to summarize the interest of this review clearly. It is just a brief comment; the review needs to emphasize the novelty of this manuscript compared to previously published reports. The number of relevant papers on melatonin as an immune enhancing/immunomodulatory agent in cancer rises into the hundreds.
Response 2: Agree. We have, accordingly, improved the conclusions to emphasize the novelty. Our changes are marked in red on page 15, but we have copied below in the particular comment all the new conclusions.
Comments 3: About the site of melatonin synthesis. The pineal gland primarily produces melatonin. Although several studies suggest that mitochondria may serve as sites of melatonin synthesis, it remains to be determined whether mitochondria constitute the primary site of melatonin production. The text would benefit from a more detailed, up-to-date discussion of this point.
Response 3: We agree about this controversial topic. Mitochondria count with the access to the precursors and the enzymes to synthesize melatonin, but only two works have addressed this specifically. He C. and collaborators tested melatonin synthesis in the mitochondria from oocytes (DOI: 10.3390/ijms17060939). Suofu et al. tested its production in the brain mitochondria (DOI: 10.1073/pnas.1705768114). We couldn’t find any more recent discoveries, only wonderful reviews that go back to these two papers. As such, we include the information from both manuscripts from line 42 (in red). Thank you so much for the suggestion.
Comments 4: Mitochondrial melatonin (lines 106–107) The statement regarding mitochondrial melatonin is controversial. Is melatonin synthesized within mitochondria, or is it synthesized elsewhere and subsequently transported into mitochondria? Please clarify the current evidence and, if applicable, highlight the main hypotheses and limitations.
Response 4: Indeed, you are right again; we assumed the mitochondrial synthesis of melatonin. We have clarified that in the text, from line 122, marked in red. Thank you so much for spotting it.
Comments 5: Melatonin and apoptosis (lines 119–120) The relationship between melatonin and apoptosis requires clarification. The manuscript suggests that melatonin inhibits apoptosis, but in cancer cells, melatonin has frequently been described as pro-apoptotic. How can these apparently opposite effects be merged? Please discuss the context-dependence of melatonin’s actions (e.g. normal vs tumour cells, dosing, timing).
Response 5: The anti-apoptotic and pro-apoptotic roles of melatonin are context-dependent. We have clarified that in line 136 (marked in red). We have paid attention to clarify it in the rest of the text when appropriate. Furthermore, we found several lines of evidence indicating that melatonin can function as an anti-apoptotic molecule. One protective mechanism involves the mitochondria: when melatonin binds to the mitochondrial MT1 receptor, it activates Gαi, which suppresses adenylate cyclase activity and prevents stress-induced cytochrome c release, thereby attenuating caspase activation (DOI: 10.1073/pnas.1705768114). Consistent with this, studies on reproductive toxicity have shown that melatonin pretreatment reduces tramadol-induced apoptosis during the pathogenesis of testicular injury (DOI: 10.22122/ahj.v12i2.265). Additional evidence reported that melatonin administration increases the anti-apoptotic protein Bcl-2 without altering levels of the pro-apoptotic protein Bax (DOI: 10.1034/j.1600-079x.2002.01891.x). This shift in the Bcl-2/Bax balance limits apoptosome formation and reduces caspase-3 activation, a key executor caspase in the apoptotic cascade (DOI: 10.1038/sj.cdd.4400476 ).
Comments 6: Disparity in effects (lines 225–226) It would be beneficial to explain the biological basis of the reported disparity in melatonin’s effects. In addition, are there other hormones or tissue-derived factors that display a similar dual or context-dependent behaviour? Providing such examples would help readers better understand this controversial aspect of melatonin’s activity.
Response 6: We have rewritten that last paragraph to favor the interpretation (from line 239 in red). Glucocorticoids and some sexual hormones (like estrogens or testosterone) can have similar behavior.
Comments 7: Nuclear receptors and lipoxygenases (line 245). The role of melatonin acting through nuclear receptors was discarded many years ago, and the published article on this point is controversial. Please reconsider the emphasis placed on nuclear receptors and discuss alternative, better-supported mechanisms by which melatonin may modulate lipoxygenase activity.
Response 7: Indeed, melatonin's role through nuclear receptors is in doubt (doi: 10.3390/molecules26092693). We have reviewed that part of the manuscript. We have now included some information about melatonin binding to calmodulin and boosting lipoxygenase activity (https://doi.org/10.1016/j.taap.2009.05.011). Changes from line 256 in red. Thank you so much for spotting that.
Comments 8: Time-dependent effects (line 255). The mechanism underlying the reported time-dependent effect of melatonin should be commented. Is there any mechanistic hypothesis or experimental evidence that could explain this temporal dependence?
Response 8: Thank you for this comment. We have added the effectors behind these changes, from line 268 in red. Indeed, the mechanism behind this behavior is the activation of downstream effectors of p38, TLR3, NF-кB…
Comments 9: Novel antitumor agent” (line 344). The sentence stating that “melatonin has been proposed as a novel antitumor agent…” is historically inaccurate. Melatonin was proposed as an antitumor agent as early as the 1960s, and therefore, it cannot be considered “novel” in this regard. Please rephrase this sentence to reflect the historical context more accurately
Response 9: Thank you so much for spotting this out. We have rephrased that sentence, which is now 373.
Comments 10: Melatonin synthesis and PKD (line 376). The statement that melatonin synthesis is compromised while, at the same time, it restores PKD is not clear. Please provide more details on the mechanism involved and discuss the possible interaction between impaired melatonin synthesis and restoration of PKD function.
Response 10: We have rephrased and completed that paragraph, so its interpretation is clearer. Thank you so much. The changes are from line 415 in red.
Comments 11: Dose, timing, and chronobiology (lines 381–382). How much melatonin was used in the cited studies, and at what time of day was it administered? Melatonin is commonly given in the late afternoon or evening, before bedtime. Does chronobiology (administration time and circadian phase) influence its efficacy as an adjuvant in cancer therapy? A brief discussion of this aspect would considerably enrich the review.
Response 11: We have checked in the original paper the strategy of melatonin administration, and they only inform about the dose, without mentioning the time of day. Which will surely have an impact. We have included the data we found, and it is included from line 424 (in red). It is a pity they don’t mention more, so we could discuss it deeper. Thank you for the suggestion!
Comments 12: HIF-1α and hypoxia (line 386). The sentence is difficult to understand as written. It seems to suggest that melatonin reduces HIF-1α activation but not hypoxia in the tumour core. Please rephrase and clarify the mechanism proposed and how melatonin might affect HIF-1α signalling and modify tumour hypoxia.
Response 12: We have rephrased that paragraph, as the mechanism is very similar to what was described in the previous comment (10), which can be found from line 429. We hope it is clearer. Thank you so much for the suggestion.
Comments 13: Lack of effect in early-stage breast cancer (line 391). The absence of an apparent beneficial effect of melatonin in randomized controlled trials in early-stage breast cancer or ductal carcinoma in situ is an important point. A brief discussion of these negative or neutral results would be highly relevant for this review, as it provides a more balanced and realistic view of melatonin’s translational potential.
Response 13: Thank you for the suggestion. It is very important to mention that in this kind of study. The time of administration can have a tremendous impact on the efficacy of melatonin. We have included that from line 407.
Comments 14: “Reduce the TME” (line 461). The sentence “This will be useful to reduce the TME” is unclear, since “reducing the TME” has no specific biological meaning. Please reformulate to indicate whether you refer to reducing immunosuppression, angiogenesis, stromal content, inflammation, or other particular features of the tumour microenvironment.
Response 14: Thank you so much. It is now rewritten; it goes from line 514.
Comments 15: “Melatonin-deficient macrophages” (line 519). The term “melatonin-deficient macrophages” is confusing. Do you mean macrophages from melatonin-deficient mice, or macrophages exposed to melatonin-deficient conditions? Please clarify the experimental model and rephrase accordingly.
Response 15: Indeed, it is confusing. This term was used by the authors. In fact, they assume the macrophages without a melatonin treatment are deficient in melatonin, which is not strictly true. Thus, we have rephrased that part to something with more sense (we hope). It starts in line 573. Thank you so much for this comment.
Comments 16: Conclusions
The Conclusions section should be strengthened to summarize the specific interest and novelty of this review clearly. It would be helpful to indicate: (i) what this review adds to the existing extensive literature on melatonin and cancer; (ii) the main conceptual advances regarding melatonin and immune modulation in the TME; and (iii) the key unresolved questions and future directions in the field.
Response 16: Thank you so much for these suggestions. The Conclusions section has been expanded and restructured to clearly articulate the novelty of this review, summarize the main conceptual advances regarding melatonin-driven immune modulation in the tumor microenvironment, and highlight key unresolved questions and future research directions. It reads like this:
This review provides an integrative perspective on melatonin that extends beyond its well-established antioxidant properties. This indoleamine acts as a central regulator of immunometabolic processes within the TME. The present work adds conceptual novelty by linking mitochondrial regulation, redox homeostasis, immune modulation, and tumor metabolism within a unified framework. A key advance highlighted here is the role of melatonin as a modulator of immune cell function within the TME, particularly through its capacity to influence TAMs polarization and immune evasion mechanisms. By preserving mitochondrial function and reprogramming cellular metabolism, melatonin promotes antitumoral immune responses while counteracting the metabolic and inflammatory conditions that favor tumor progression. This positions melatonin not merely as an adjuvant antioxidant, but as an active modulator of the tumor immune landscape. Despite these promising insights, several unresolved questions remain. The context-dependent effects of melatonin across different tumor types, stages, and immune microenvironments require further clarification. In addition, optimal dosing regimens, chronobiological considerations, and potential synergies between melatonin and immunotherapies or metabolic-targeting agents warrant systematic investigation. Addressing these challenges will be essential to translate the immunometabolic properties of melatonin into effective therapeutic strategies.
Overall, this review underscores the emerging concept of melatonin as a modulator at the intersection of mitochondrial function, immune regulation, and tumor biology, offering a coherent framework to guide future experimental and clinical studies.

Reviewer 2 Report
The article comprehensively discusses melatonin's antioxidant effects, immune modulation, macrophage polarization, and its role in cancer therapy. The literature is extensive, the topic is current, and the topic addresses a particularly important area of immuno-oncology.
However, in its current form, the study is overly broad, the flow between chapters is interrupted, some sections are repetitive, the explanations for schematic figures are inadequate (e.g., Figures 1–3; missing explanations in relevant contexts), and some critical mechanisms remain superficial.
The article has significant potential but requires extensive revision to become publishable.
Structural integrity issue:
The article covers a very broad area; topics such as oxidative stress, mitochondrial function, immunity, cancer, TAMs, metabolic reprogramming, etc., are presented sequentially but disjointedly. Transitions between sections are weak.
Source imbalance (mixed very old / very new sources).
Lack of mechanistic depth:
Specifically:
NF-κB / SIRT3 / MT1/MT2 receptor interactions
TAM metabolic reprogramming
melatonin immunometabolic pathways
Linking the ROS–mitochondria–apoptosis triangle
are inadequately explained.
The study's key claims need to be supported by a more robust molecular basis.
The cancer section is very superficial:
Tumor microscopy, TAM subtypes, metabolic reprogramming, and immune evasion mechanisms are only briefly covered.
Originality issue: The review is very comprehensive, but comparative analyses that would offer a new perspective are lacking.
Author Response
Thank you very much for taking the time to review this manuscript. We really appreciate all your comments, we believe they truly improved the quality of our review. We will answer your comments in detail below:
However, in its current form, the study is overly broad, the flow between chapters is interrupted, some sections are repetitive, the explanations for schematic figures are inadequate (e.g., Figures 1–3; missing explanations in relevant contexts), and some critical mechanisms remain superficial.
Response 1: Thanks for the suggestions. We have included new paragraphs to improve the flow between sections. We have improved explanations for our figures and increased the mechanistic depth of the aspects mentioned below. A native friend has reviewed all the English and spotted the not that English sentences, which are rewritten (also in red).
Structural integrity issue:
The article covers a very broad area; topics such as oxidative stress, mitochondrial function, immunity, cancer, TAMs, metabolic reprogramming, etc., are presented sequentially but disjointedly. Transitions between sections are weak.
Response 2: Thank you so much for this suggestion. We have strengthened the structural coherence of the manuscript by introducing explicit transition paragraphs between sections, clarifying how oxidative stress, mitochondrial function, immune regulation, and cancer biology are mechanistically interconnected and converge within the tumor microenvironment.
Source imbalance (mixed very old / very new sources).
Response 3: We have realized that we have some very old bibliography, in some cases. This is due to the relevance of those papers when they described for the first time the qualities of melatonin. We work from those aspects now, and we should recognize their value. To create a better balance, we have also included more new references.
Lack of mechanistic depth:
Specifically:
NF-κB / SIRT3 / MT1/MT2 receptor interactions
Response 4: Thank you so much for pointing this out. We have improved the mechanistic information in the NF-κB / SIRT3 / MT1/MT2 receptor interactions. We have included that information from line 109.
TAM metabolic reprogramming
Response 5: Thank you so much for the suggestion. We have included more mechanistic information. From line 588 labelled in red.
melatonin immunometabolic pathways
Response 6: We have included changes regarding this aspect in different sections:
1.- Metabolic changes in tumoral cells when treated with melatonin. This metabolic reprogramming affects mainly the Warburg effect, or how pyruvate is transformed into Acetyl-CoA after melatonin treatment. From line 415 (changes in red).
2.- Melatonin rewires glycolysis to reduce the acidic environment and activates T-cells. From line 498, in red.
3.- Changes from line 590 will also add mechanistic depth in this part.
Linking the ROS–mitochondria–apoptosis triangle
Response 7: Indeed, we have included that triangle, as shown from line 136. Thank you for spotting it.
The study's key claims need to be supported by a more robust molecular basis.
The cancer section is very superficial:
Tumor microscopy, TAM subtypes, metabolic reprogramming, and immune evasion mechanisms are only briefly covered.
Response 8: Thank you so much for your suggestion. We have expanded the cancer section to provide a deeper mechanistic discussion on tumor metabolic heterogeneity, TAM subtypes, and immune evasion mechanisms, emphasizing how melatonin integrates metabolic and immune regulation within the tumor microenvironment
Originality issue: The review is very comprehensive, but comparative analyses that would offer a new perspective are lacking.
Response 9: Thanks for your kind words. To enhance the originality of the review, we have incorporated a comparative analysis positioning melatonin in relation to classical antioxidants, immunomodulators, and metabolic regulators, highlighting its unique mitochondrial, immunometabolic, and circadian-integrative properties

Reviewer 3 Report
The authors summarized the bioactive functions of melatonin in maintaining health. They collected information showing that melatonin eliminates free radicals by directly scavenging reactive oxygen and nitrogen species (ROS/RNS) and by upregulating key antioxidant enzymes such as superoxide dismutase (SOD) and catalase (CAT). It is considered significantly more effective than classical antioxidants and protects mitochondrial function, improving oxidative phosphorylation. The authors also describe the wide-ranging immunomodulatory roles of melatonin, affecting both innate and adaptive immune responses. These actions converge in the tumor microenvironment, where melatonin promotes anti-tumor M1 macrophage phenotypes over pro-tumoral M2 states. In addition, the authors show that melatonin exhibits multiple anti-cancer effects by suppressing angiogenesis, inhibiting EMT, reversing the Warburg effect, and downregulating PD-L1 expression, thereby enhancing antitumor immunity. Overall, this profile highlights the potential of melatonin as an adjuvant in cancer prevention and treatment.
I have some suggestions for the authors to improve the manuscript. It would be helpful to show whether melatonin receptors have cell-specific expression patterns, since receptor distribution is the first factor that determines how cells respond. Summarizing this information together in Figure 1 would make the presentation clearer. Because melatonin can be produced endogenously, it would also be useful if the authors could provide information related to factors that cause melatonin levels to decrease.
Author Response
I have some suggestions for the authors to improve the manuscript.
Thank you so much for your comments and your suggestions to improve the manuscript. They are very valuable. We have tried to address the best of our capabilities, we hope you think it's ok.
Comments 1: It would be helpful to show whether melatonin receptors have cell-specific expression patterns, since receptor distribution is the first factor that determines how cells respond. Summarizing this information together in Figure 1 would make the presentation clearer.
Response 1: Thank you so much for this suggestion. We have underlined these differences (from line 74 labelled in red), and in Figure 1.
Comments 2: Because melatonin can be produced endogenously, it would also be useful if the authors could provide information related to factors that cause melatonin levels to decrease.
Response 2: Indeed, there was some important bit of information that we have not included. Thank you so much for the suggestion, now it is written from line 57 in red.

Round 2
Reviewer 2 Report
The authors made the corrections I requested. Thank you.
The authors made the corrections I requested. Thank you.
Reviewer 3 Report
The authors had included my suggestions to improve the manuscript. I agree that this manuscript meets the criteria for publication.
The authors had included my suggestions to improve the manuscript. I agree that this manuscript meets the criteria for publication.